# Prolonged maternal investment in northern bottlenose whales alters our understanding of beaked whale reproductive life history

**Laura Joan Feyrer**[1]*, **Shu ting Zhao**[2], **Hal Whitehead**[1], **Cory J. D. Matthews**[2]

**1** Department of Biology, Dalhousie University, Halifax, Nova Scotia, Canada, **2** Arctic Aquatic Research Division, Fisheries and Oceans Canada, Winnipeg, Canada

* ljfeyrer@dal.ca

**Data Availability Statement:** Stable isotope data used in this study is available from the dryad database (DOI: https://doi.org/10.5061/dryad. k98sf7m3j).

## Abstract

Nursing and weaning periods are poorly understood in cetaceans due to the difficulty of assessing underwater behaviour in the wild. However, the onset and completion of weaning are critical turning points for individual development and survival, with implications for a species' life history including reproductive potential. $\delta^{15}N$ and $\delta^{13}C$ deposited in odontocete teeth annuli provide a lifetime record of diet, offering an opportunity to investigate variation and trends in fundamental biology. While available reproductive parameters for beaked whales have largely been inferred from single records of stranded or hunted animals and extrapolated across species, here we examine the weaning strategy and nursing duration in northern bottlenose whales (*Hyperoodon ampullatus*) by measuring stable isotopes deposited in dentine growth layer groups (GLGs). Using a collection of *H. ampullatus* teeth taken from whales killed during the whaling era (N = 48) and from two stranded specimens, we compared ontogenetic variation of $\delta^{15}N$ and $\delta^{13}C$ found in annual GLGs across all individuals, by sex and by region. We detected age-based trends in both $\delta^{15}N$ and $\delta^{13}C$ that are consistent across regions and males and females, and indicate that nursing is prolonged and weaning does not conclude until whales are 3–4 years old, substantially later than previous estimates of 1 year. Incorporating a prolonged period of maternal care into *H. ampullatus* life history significantly reduces their reproductive potential, with broad implications for models of beaked whale life history, energetics and the species' recovery from whaling.

## Introduction

Maternal investment in mammals varies based on an array of ecological and evolutionary factors resulting in a range of maternal strategies e.g. [1]. Nursing is critical to the survival and fitness of infant mammals; providing our earliest energetic and nutritional requirements, supporting maternal bonding, and initializing ongoing socialization [2–4]. Nursing duration and the weaning strategy have implications for infant survival, interbirth interval, and lifetime reproductive output, which are critical measures for understanding the life history, energetics and population dynamics of a species [5]. While lactation may occur over a period of weeks to

**Funding:** LJF, HW and CJDM received funding from Fisheries and Oceans Species at Risk program (https://www.dfo-mpo.gc.ca/species-especes/sara-lep/index-eng.html) for this work. In addition, LJF received funding from the Natural Sciences and Engineering Research Council of Canada and Killam Trusts. The funders had no role in study design, data collection and analysis, decision to publish, or preparation of the manuscript.

**Competing interests:** The authors have declared that no competing interests exist.

years, weaning initiation and completion are important developmental turning points–as juveniles become nutritionally independent, it allows females to redirect significant energetic resources back to themselves and towards their future offspring [6,7]. Weaning, which may be sudden or gradual, depends on a range of factors including the survival and vulnerability of offspring in the postpartum period, the technical difficulty of self-sufficient foraging strategies, species social structure, individual behavioural plasticity and regional prey availability [3,5,8,9]. Responding to a range of ecological and evolutionary factors, nursing duration can vary widely among and even within species, forming the context of the weaning "conflict", with trade-offs between the fitness of offspring and future female reproductive potential [3,4,10].

Maternal investment in cetaceans (dolphins and whales) is known to be extensive and provides a key role in infant survival, however our appreciation of weaning strategies is challenged by the cryptic nature of nursing behaviour and their aquatic habitat [1]. What we do know can be generalized by sub-order; with a large degree of variability between species, odontocetes appear to prolong nursing and weaning over years (mean = 21 months), while mysticetes typically wean their young within the first year (mean = 11 months) [11] (S1 Table). This difference in maternal investment has been linked to energetic resources available to income versus capital breeders [1,12].

Four different methods have been used to estimate nursing duration in cetaceans: stomach content analysis, cow-calf ratios, behavioural observations and stable isotope analyses, which may explain some of the discrepancies between estimates within and among species [1,11]. Across studies, behavioural observations typically reported the oldest average age at weaning (27 months), in contrast with stomach content analyses, which found average weaning age occurred much younger (16 months, S1 Table). Temporal analysis of nitrogen stable isotopes ($\delta^{15}N$) in accretionary tissues, such as sequential growth layer groups (GLGs) in dentine, have also been used to estimate weaning age and other ontogenetic shifts in individual foraging and trophic level based on nutritional physiology [13–17]. As $\delta^{15}N$ decreases during the transition from juveniles feeding exclusively on milk to independent foraging, differences in $\delta^{15}N$ between GLGs in tooth dentine can be used to estimate nursing duration and weaning completion e.g. [14,16]. However weaning related relationships with $\delta^{13}C$ are less clear and across studies there is no consistent trend or pattern reported for isotopic carbon found in marine mammal tissues during the dietary transition from milk to prey (e.g. [16–18]).

Due to the offshore habitat and elusive nature of deep diving beaked whales (Ziphiidae), there is a lack of baseline data on key aspects of their life history so that reproductive parameters are poorly understood [19]. Much of our understanding of their biology comes from one species, the northern bottlenose whale (*Hyperoodon ampullatus*), which was the target of a century of commercial whaling across the North Atlantic ending in the early 1970's. In the final years of the commercial hunt in Labrador and northern Iceland, data otherwise difficult to collect today using non-lethal methods were recorded for many individuals, including age (from teeth), sex, sexual maturity, reproductive state, fetal term, and stomach contents [20]. Whaling records for the species provide the only estimates of reproductive parameters, which have been the basis for previous studies of beaked whale energetics (e.g. [5,21]) and include: gestation (12 months)—based on fetal growth curves; lactation length (~ 1 year)—based on a single calf that had both milk and squid in its stomach; resulting in a combined estimate of calving interval (2 years), which was also supported by an accumulation of 0.5 corpora per year in mature females [20].

Similar to other odontocetes, dentine GLGs in *H. ampullatus* form annually deposited layers which have been used to age individual specimens [22,23]. However beaked whales are unique among odontocetes in that most only have a single pair of tusk-like teeth that erupt in

mature males and remain embedded in the jaw of juveniles and females [24]. Likely due to the difficulty in accessing tooth specimens, this study is the first investigation of ontogenetic diet shifts using stable isotopes for any species of beaked whale, based on samples from an unusually large collection (N = 151 individuals) of *H. ampullatus* teeth taken from whales that were commercially hunted in the North Atlantic.

Our primary objective was to characterize nursing duration and the end of the weaning period in individual *H. ampullatus* using $\delta^{15}N$ and $\delta^{13}C$, accounting for potential differences due to sex or regional variation. We test the hypothesis that nursing extends beyond one year, in contrast to Benjaminsen & Christensen's [20] inference based on stomach contents of a single calf. Similar to *Physeter macrocephalus*, another deep diving cetacean with prolonged maternal care [25], beaked whales regularly dive to extreme depths (~1000m) to feed on mesopelagic and epibenthic prey [19]. As a result, juveniles may not be physically capable of independent foraging until they have grown large enough to be competent divers or engage in demanding foraging strategies, the complexities of which are currently poorly understood. Secondarily we compare differences in diet between juveniles and adults to assess whether, similar to other odontocetes (e.g. *Orcinus orca* [14]; *P. macrocephalus* [13]), there is evidence of increases in dietary trophic level with age. This study offers a rare opportunity to expand our appreciation of the variation in maternal investment strategies in beaked whales and across cetaceans.

## Materials and methods

### Tooth collection and dentine sampling

Teeth were taken from *H. ampullatus* killed by Norwegian whalers in the waters off northern Iceland in 1967 and northern Labrador in 1971 [26] (Fig 1). Northern bottlenose whales are usually found in groups of one to four, and whalers would take all the whales they encountered, regardless of sex or age class, so we assume our dataset has low demographic capture bias [20]. Individuals included in this analysis ranged from 4–27 years old (median age = 14). The teeth of two *H. ampullatus* that stranded in northeast Newfoundland in 2004 were also analyzed. As specimens were part of an archived natural history collection, no approval from the University Committee on Laboratory Animals was required.

The jaws of whaled specimens were originally boiled for two hours to facilitate tooth extraction [26]. Teeth were sectioned along the longitudinal midline and stored unpreserved at room temperature in individual sachets for over 40 years prior to this study. Genetic analysis of gum-tissue from the teeth used in this study confirmed the sex documented in the whaling records [27,28]. The teeth from Newfoundland animals were extracted from decomposed specimens, air dried and stored whole until being sectioned for this study. Similar to other odontocetes [29–31], *H. ampullatus* dentine is laminated, with one clear and one opaque layer defining each annual GLG within the cone of the tooth [22] (Fig 2). Only teeth with a clear neo-natal line and defined GLG structure across the first five years were retained for isotope analysis, reducing our sample size to 50 individuals (N = 6 from Iceland, N = 42 from Labrador, N = 2 from Newfoundland). To improve GLG definition, tooth sections were initially polished using 30μm aluminum oxide lapping film [16] and then acid-etched using 10% formic acid [32]. GLGs were counted and aged assuming annual deposition, starting at the line that divides prenatal and postnatal dentine [16,22]. Using a single section of each tooth, GLGs 1–5 were sampled individually at a depth of 250-μm with a 300-μm-diameter drill bit, using a high-resolution micro-mill (New Wave Research, Freemont, California). When sufficient prenatal dentine was present it was sampled at a depth of 150 μm. For mature individuals (> 9 years old) [26], we also collected samples from older GLGs as a

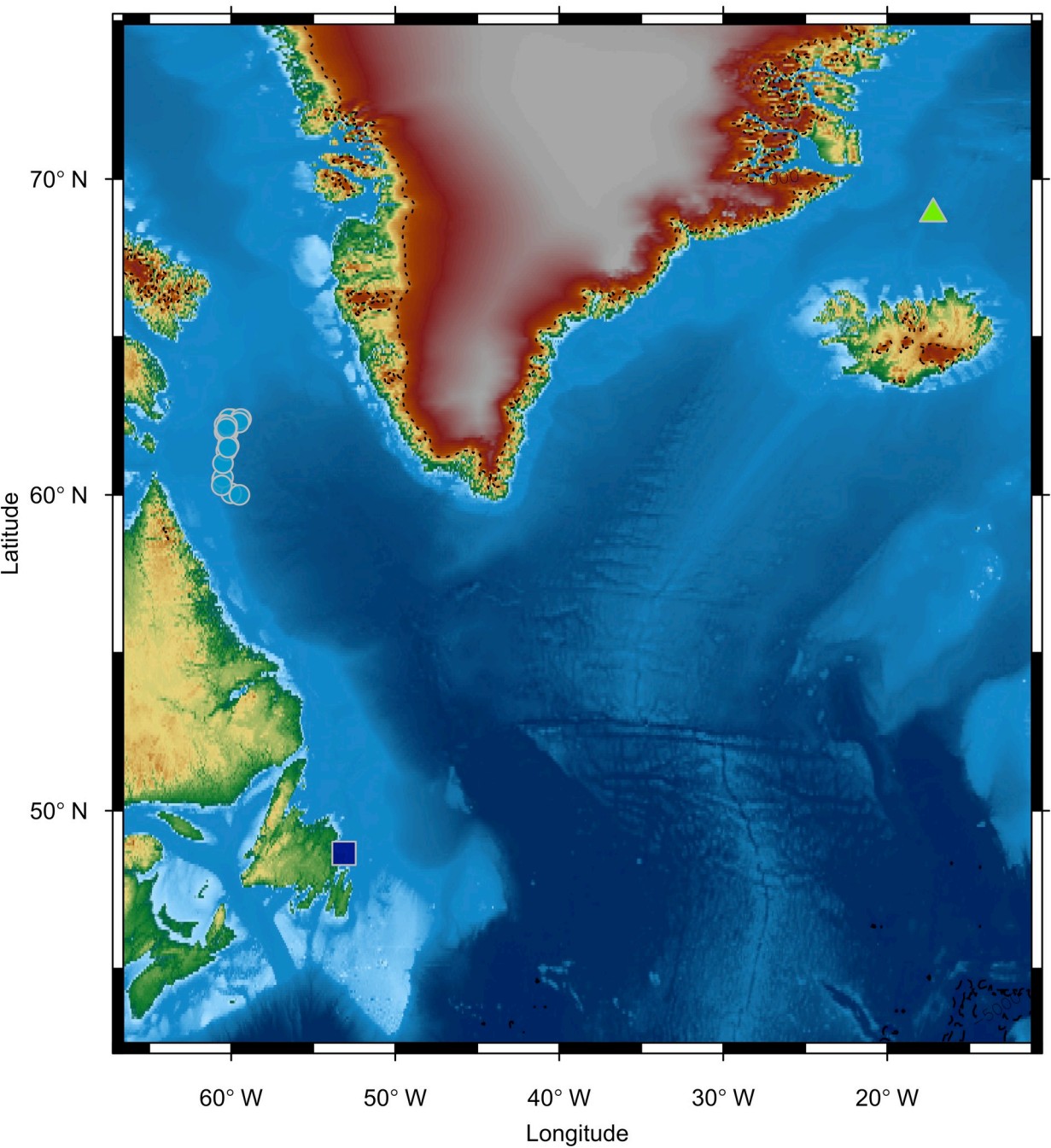

**Fig 1. Map of study area regions and specimen collection locations.** Green triangle = Iceland, light blue dots = Northern Labrador, dark blue square = Newfoundland strandings.

proxy for adult diet (N = 29). However, as whales age their GLGs become compressed and are not wide enough to sample individually. Instead we collected samples representative of the mature age class by drilling across GLGs 8–12 as a group with a 1 mm-diameter drill bit using a Dremel hand tool.

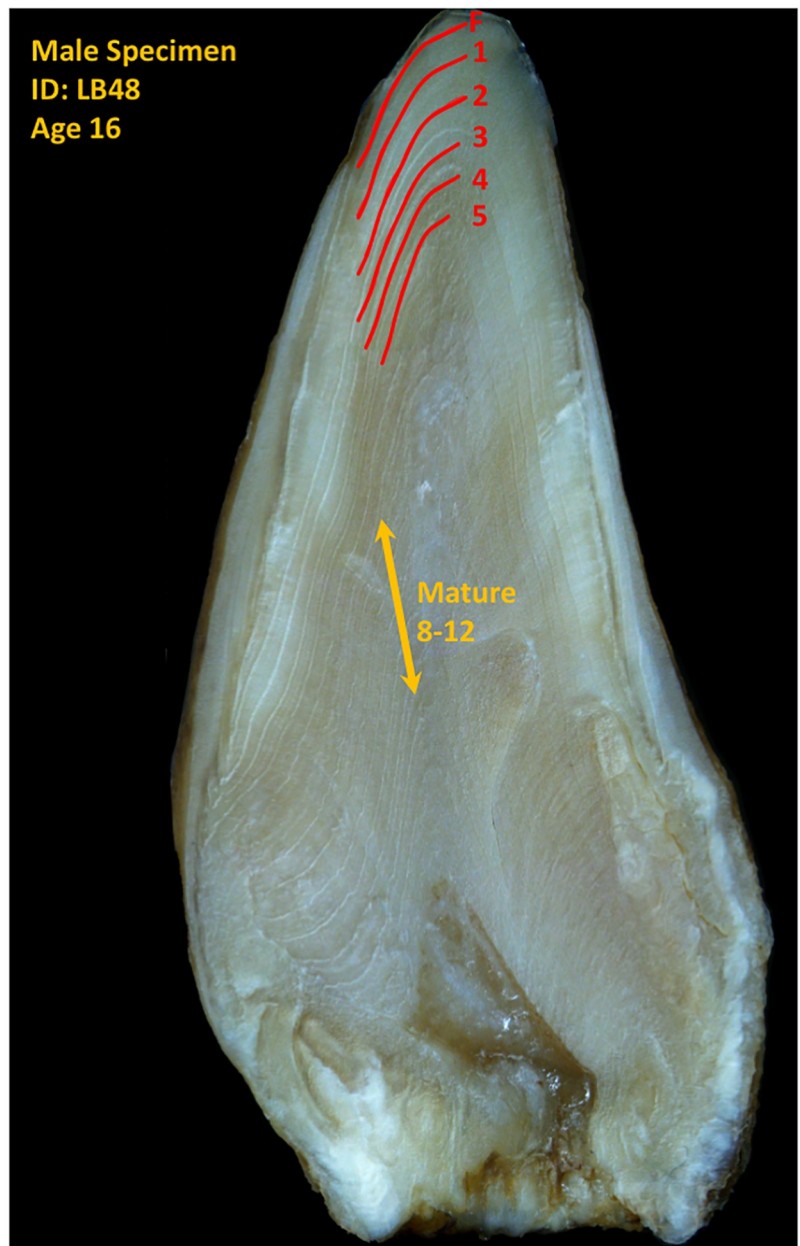

**Fig 2. A sectioned *H. ampullatus* tooth prior to sampling.** GLGs are annotated: F = fetal, 1–5 = years (red lines) and mature = sampling across years 8–12 (yellow line).

## Stable isotope analysis ($\delta^{15}$N / $\delta^{13}$C)

Powdered dentine from each sampled GLG was weighed (~1 mg) into tin cups for isotopic analysis on a Vario EL Cube elemental analyzer (Elementar, Germany) connected to a DELTA Advantage isotope ratio mass spectrometer (Thermo, Germany). Isotope ratios are reported in Delta notation ($\delta$) as per mil (‰) deviation from isotope ratios of atmospheric $N_2$ for nitrogen and Vienna Pee-Dee Belemnite (V-PDB) limestone for carbon. $\delta^{15}$N or $\delta^{13}$C are defined as $\delta = (R_{sample} - R_{standard})/R_{standard}$), where R is the ratio of the abundance of the heavy to the light

isotope. Values are normalized to internal standards nicotinamide, ammonium sulfate + sucrose, caffeine, and glutamic acid, whose isotopic compositions cover the natural range of samples ($\delta^{15}$N -16.61 to 16.58‰, $\delta^{13}$C -34.46 to –11.94‰) and are calibrated to international standards IAEA-N1(+0.4‰), IAEA-N2(+20.3‰), USGS-40(-4.52‰) and USGS-41(47.57‰) for $\delta^{15}$N, and IAEA-CH-6(-10.4‰), NBS-22(-29.91‰), USGS-40(-26.24‰) and USGS-41 (37.76‰) for $\delta^{13}$C. Analytical precision based on repeated measures of laboratory reference materials not used in calibrations was ~0.1‰ for both $\delta^{15}$N and $\delta^{13}$C within multiple laboratory runs. Variation between duplicate measures of ~10% of samples had an absolute mean of 0.26 ‰ for $\delta^{15}$N and 0.21 ‰ for $\delta^{13}$C.

The small size of some GLGs meant it was sometimes necessary to collect amounts less than 1 mg. A linearity study showed samples <0.5 mg appeared to have a positive bias in $\delta^{15}$N but not $\delta^{13}$C, and further analysis was restricted to samples weighing >0.5mg, reducing the number of GLG samples available for some individuals. Additionally, we omitted the smallest duplicate sample, so that only a single sample from an individual GLG was included in further analysis [33].

## Data analysis

Following the screening for duplicates and sample weight described above, 50 individuals were included in summary statistics regardless of how many GLGs were available. However, ontogenetic trend analysis was restricted to those individuals which had stable isotope data available from at least GLGs 1–3 (N = 37). Data structure, variables, and sample sizes are identified in Table 1 and variable inclusion rationale and data sources are further described in S2 Table.

For comparison with other published values and ecological studies, carbon isotope values were adjusted for the oceanic Suess effect, applying a factor of 0.0019‰ yr $^{-1}$ to $\delta^{13}$C measured in GLGs; $\delta^{13}$C$_{cor}$ values are approximately relative to the year 2000 [16,34,35]. The isotope values sampled from a cross section of mature GLGs (age 8–12) were assumed to represent the average isotopic profile of adult whales, and used as a benchmark for assessing when the weaning associated $\delta^{15}$N decline ended.

The dataset was initially summarized and explored for the presence of ontogenetic trends in nitrogen and carbon isotope ratios. The effect of sex and region on isotopic composition was initially evaluated using two-sample t-tests. A hierarchical linear mixed effects regression model implemented with the lme4 package in R (Version 3.0.1 [36]) assessed the effects of sex, region and GLG. Given uneven sample sizes between GLGs, we used a paired t-test to consider the distinction between subsequent GLGs. Due to the small sample size (N = 2) and differences in source collection from other samples, Newfoundland specimens were not included in statistical summaries or tests unless specified.

To investigate ontogenetic trends and nursing duration, for each individual with samples from GLGs 1–3 (N = 37) we calculated the ‰ difference between GLG 1 and all other available GLGs (fetal dentine, GLGs 2-maturity). Three methods of determining weaning completion

**Table 1. Data structure, variables and sample sizes.**

| Dependent variables | Independent variables | Total individuals | | GLG chronologies | |
|---|---|---|---|---|---|
| | | N = GLG samples | | N = GLG samples | |
| $\delta^{15}$N | Region | 50 IDs | | 39 IDs | |
| $\delta^{13}$C | GLG–Year, Age Class | N = 244 GLGs | | N = 207 GLGs | |
| | Sex | (*288 including duplicate samples) | | | |
| | Individual Age | | | | |

were compared for individuals which had samples collected from mature age classes by calculating the age: (A) when $\delta^{15}N$ values stopped decreasing (e.g. the lowest value of $\delta^{15}N$ in the chronology, [18]; (B) when $\delta^{15}N$ was equal to the value for their mature age class value (+/- 0.25 ‰) [14]; and (C) when $\delta^{15}N$ was -1.2‰ lower than GLG1(+- 0.25 ‰) [14,16]. The threshold for (C) was based on an average ‰ difference between GLG1 and mature samples in this study, and similar differences found in other studies of weaning in odontocetes [14,16]. For each method, individual age at weaning completion was compared by sex and between Labrador and Iceland regions using a two-sample t-test. Small sample size for Newfoundland precluded inclusion in significance tests.

## Results

### Nitrogen

Across individual chronologies, we found $\delta^{15}N$ generally peaked in GLG1 (mean = 17.73, SE = 0.10) and then declined with age. Within individuals, the relative decline in $\delta^{15}N$ between GLG 1 and all other GLG years averaged– 1.02 ‰ (Fig 3a). GLG 1 $\delta^{15}N$ was higher (mean = 0.93 ‰) than fetal dentine (mean = 17.00, SE = 0.16) and 1.06 ‰ higher than mature age class values (mean = 16.62, SE = 0.09). $\delta^{15}N$ values across all GLGs from Labrador and Iceland ranged ~3.8 ‰ (15.16 to 19.0‰). For the two specimens from Newfoundland, $\delta^{15}N$ spanned 4.8‰ and was lower (range 12.9–17.7‰) than average values from Labrador and

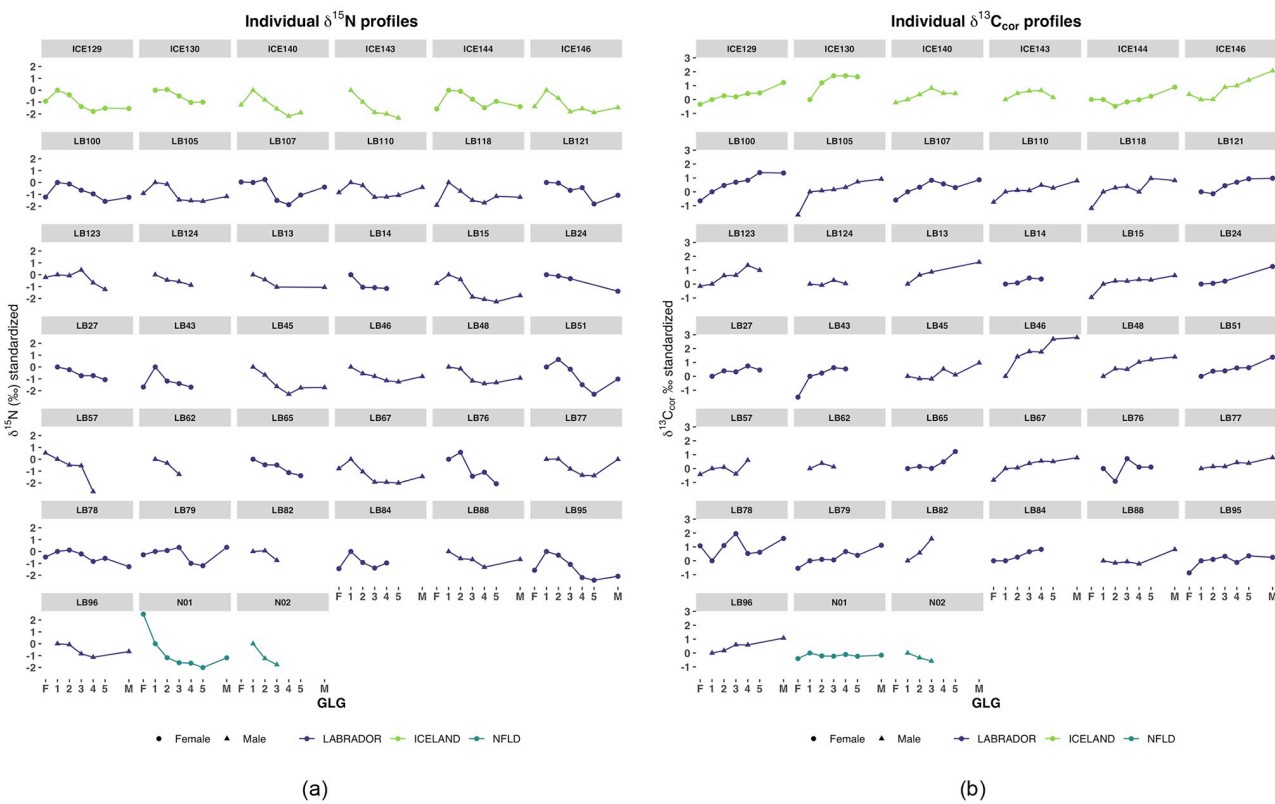

**Fig 3. Individual chronologies for (a) δ $^{15}$N and (b) δ $^{13}$C for each region.** Regions are indicated by colour. Isotope values were standardized to be relative to GLG 1 for prior (fetal dentine = F) or subsequent (years 2—mature = M) GLGs. Sex of specimen is indicated by circle (female) and triangles (male).

Iceland. For GLGs > 1, both Newfoundland specimens were greater than 1 standard deviation lower in $\delta^{15}N$ than other regions, with the adult female ~3‰ lower across GLGs.

## Carbon

$\delta^{13}C_{cor}$ values generally increased with age (mean increase in $\delta^{13}C_{cor}$ per GLG = 0.37, fetal to maturity). GLG1 was on average more enriched in $^{13}C_{cor}$ (+0.51 ‰, mean = -14.07) than fetal dentine (mean = -14.48), and more depleted than older GLGs. The range of $\delta^{13}C_{cor}$ values for mature samples were on average 1.06 ‰ higher than GLG 1 (Fig 3b). Between regions, $\delta^{13}C_{cor}$ in Labrador and Iceland were higher (-15.66 to -12.57 ‰) than Newfoundland (-17.17 to—14.78‰). The juvenile male whale from Newfoundland was one notable exception to the overall ontogenetic increase in carbon, as his $\delta^{13}C_{cor}$ values declined from GLG 1 to 3 (Fig 3b). Across all GLGs, $\delta^{13}C_{cor}$ values for the Newfoundland specimens were > 1 standard deviation below Labrador or Iceland specimen GLGs.

## Influence of sex, region and GLG

Average values of $\delta^{15}N$ and $\delta^{13}C_{cor}$ for females and males had considerable overlap and did not demonstrate a consistent pattern or significant difference between sexes across GLG's (Fig 4a and 4b, $t = 0.85$, df = 53.2, $p = 0.39$). Differences in values of $\delta^{15}N$ and $\delta^{13}C_{cor}$ between Labrador and Iceland were not significant (Fig 5a and 5b, $t = 0.58$, df = 12.1, $p = 0.57$).

Mixed effects models, implementing individual as a random effect, compared 8 different combinations of fixed effects including GLG, Region and Sex (Table 2a and 2b). Only Region and GLG were retained in the best fit mixed effect models for predicting relative $\delta^{15}N$ and $\delta^{13}C$ values. Model fit, assessed using $\Delta$ AIC $\leqq$ 2, indicated GLG was important for explaining both $\delta^{15}N$ and $\delta^{13}C$, Region was included in all best fit models for $\delta^{15}N$ and in one model for $\delta^{13}C$, Sex was also included in one of the best models for $\delta^{13}C$ (Table 2). Given the overlap in mean values with standard error between Labrador and Iceland and between males and females we conclude that the influence of region and sex on isotopic profiles are small relative to the variation attributed to GLG (age) and individual.

Paired t-tests assessing the difference between $\delta^{15}N$ and $\delta^{13}C$ of an individual between consecutive GLG's found significant differences between $\delta^{15}N$ in GLG pairs 1 through 4 and between $\delta^{13}C$ in GLG pairs Fetal (F) through age 3 (Table 3). GLGs 5 and mature (M) were also significantly different for both isotopes.

## Weaning completion

Nursing duration ranged across methods with median age of three to five. Method (A) provided older estimates of weaning completion (mean = 4.5), while methods (B) and (C) suggested weaning was completed earlier, with mean ages of 3.4. There was no substantial difference in nursing duration between Labrador or Iceland regions or with sex (Table 4) using any of the weaning analysis methods.

## Discussion

We conclude that *H. ampullatus* have a prolonged nursing period, based on a slow decrease in $\delta^{15}N$ over GLGs 1–5. This decline was generally consistent across regions (N = 50 individuals) and between sexes (N = 48 individuals) and based on a chronological analysis of 39 individuals we found that weaning ends on average between ages three and four. Extended maternal care has not previously been documented in a beaked whale species and is in contrast to the only other estimate for *H. ampullatus* completing nursing in their first year, which was based on the

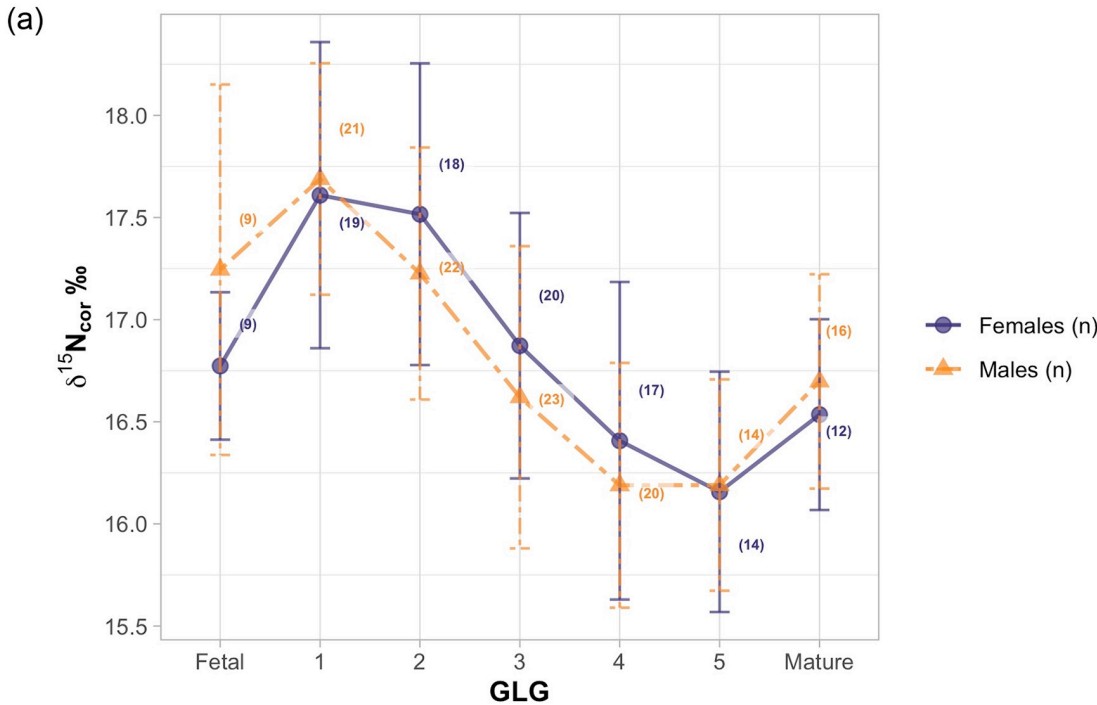

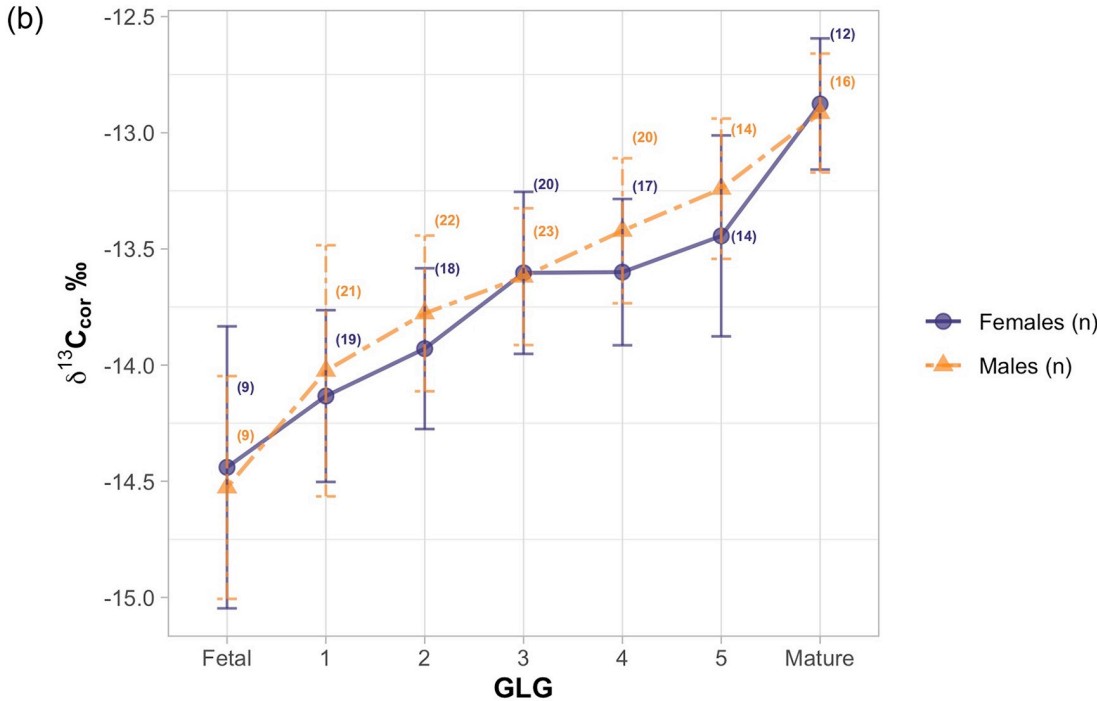

**Fig 4. Ontogenetic trends in average (a) $\delta^{15}N$ and (b) $\delta^{13}C$ by sex.** Females (N = 109 GLG samples) are purple points and males (N = 125 GLG samples) are orange triangles. Whisker bars represent standard deviation. Iceland and Labrador samples only.

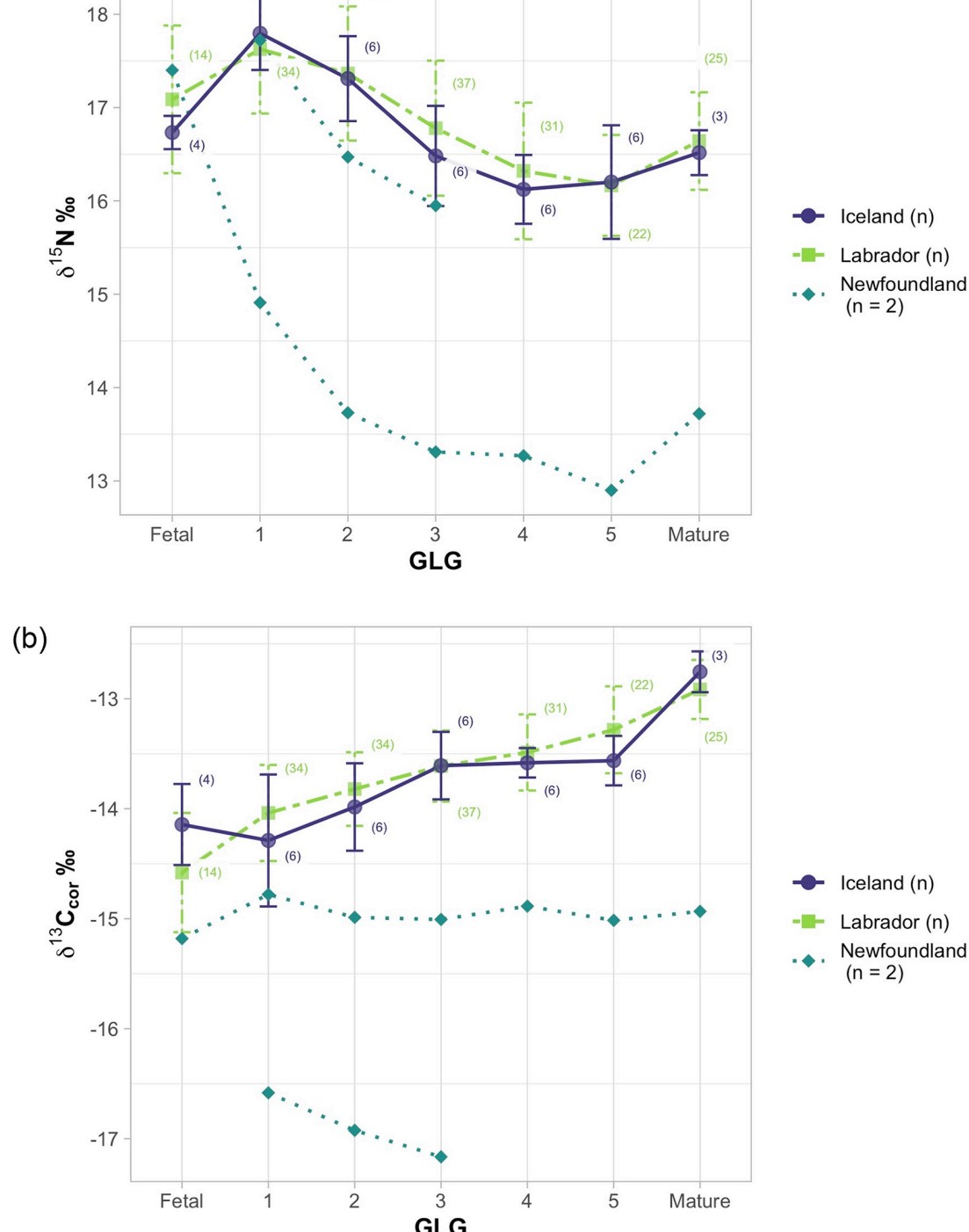

**Fig 5. Ontogenetic trends by region for values of (a) $\delta^{15}$N and (b) $\delta^{13}$C.** Purple points and green squares are mean values with standard deviation for Labrador and Iceland, blue diamonds are individual values of two specimens from Newfoundland.

**Table 2. Mixed effect model results comparisons for (a) $\delta^{15}$N and (b) $\delta^{13}$C.** Best fit models are indicated in bold based on lowest AIC score and $\Delta$ AIC $\leqq$ 2. BIC and Log Likelihood (logLik) scores with degrees of freedom (df) are included for comparison. "(1|ID)" indicates an individual effect.

**(a)**

| $\delta^{15}$N | $\Delta$AIC | AIC | BIC | logLik | df |
|---|---|---|---|---|---|
| **~GLG_N+Region +(1\|ID)** | **0.0** | **329.3** | **345.1** | **-159.6** | **5** |
| **~Sex +GLG_N+Region +(1\|ID)** | **0.5** | **329.8** | **348.7** | **-158.9** | **6** |
| **~GLG_N\*Region +(1\|ID)** | **1.5** | **331.3** | **350.2** | **-159.6** | **6** |
| ~GLG_N +(1\|ID) | 2.3 | 331.6 | 344.2 | -161.8 | 4 |
| ~Sex\*GLG_N+Region +(1\|ID) | 2.4 | 331.6 | 353.8 | -158.8 | 7 |
| ~Sex\*GLG_N+(1\|ID) | 2.7 | 334.3 | 353.3 | -161.1 | 6 |
| ~Sex+GLG_N +(1\|ID) | 3.2 | 332.5 | 348.3 | -161.2 | 5 |
| ~Sex+Region +(1\|ID) | 25.1 | 354.3 | 370.1 | -172.2 | 5 |
| ~1 +(1\|ID) | 26.9 | 356.2 | 365.7 | -175.1 | 3 |

**(b)**

| $\delta^{13}$C | $\Delta$AIC | AIC | BIC | logLik | df |
|---|---|---|---|---|---|
| **~GLG_N +(1\|ID)** | **0.0** | **198.9** | **211.6** | **-95.5** | **4** |
| **~GLG_N+Region +(1\|ID)** | **1.0** | **199.9** | **215.7** | **-94.9** | **5** |
| **~Sex+GLG_N +(1\|ID)** | **2.0** | **200.9** | **216.7** | **-95.4** | **5** |
| ~GLG_N\*Region +(1\|ID) | 2.2 | 201.1 | 220.0 | -94.5 | 6 |
| ~Sex\*GLG_N +(1\|ID) | 2.7 | 201.6 | 220.6 | -94.8 | 6 |
| ~Sex +GLG_N +Region+(1\|ID) | 2.9 | 201.8 | 220.8 | -94.9 | 6 |
| ~Sex\*GLG_N +Region+(1\|ID) | 3.6 | 202.5 | 224.6 | -94.3 | 7 |
| ~1 +(1\|ID) | 135.8 | 334.7 | 344.2 | -164.3 | 3 |
| ~Sex + Region + (1\|ID) | 139.0 | 337.9 | 353.7 | -164.0 | 5 |

**Table 3. Paired t-test results for comparisons between GLG years within individuals for (a) $\delta^{15}$N and (b) $\delta^{13}$C.** Test significance (p-value), mean difference in ‰ (Mean dif. ‰), confidence intervals of the difference (C.I. ‰) and degrees of freedom (df) are presented for each test.

**(a)**

**$\delta^{15}$N**

| GLG | p-value | Mean dif. (‰) | C.I. (‰) | | df |
|---|---|---|---|---|---|
| **F to 1** | 0.005 | -0.72 | -1.23 | -0.26 | 18 |
| **1 to 2** | <0.001 | 0.37 | 0.22 | 0.52 | 38 |
| **2 to 3** | <0.001 | 0.65 | 0.50 | 0.81 | 40 |
| **3 to 4** | <0.001 | 0.40 | 0.24 | 0.57 | 37 |
| **4 to 5** | 0.322 | 0.09 | -0.10 | 0.28 | 28 |
| **5 to M** | 0.001 | -0.49 | -0.74 | -0.23 | 20 |

**(b)**

**$\delta^{13}$C**

| GLG | p-value | Mean dif. (‰) | CI (‰) | | df |
|---|---|---|---|---|---|
| **F to 1** | 0.003 | -0.51 | -0.81 | -0.20 | 18 |
| **1 to 2** | 0.002 | -0.22 | -0.36 | -0.09 | 38 |
| **2 to 3** | <0.001 | -0.24 | -0.36 | -0.12 | 40 |
| **3 to 4** | 0.225 | -0.09 | -0.23 | 0.06 | 37 |
| **4 to 5** | 0.086 | -0.13 | -0.28 | 0.02 | 28 |
| **5 to M** | <0.001 | -0.35 | -0.53 | -0.17 | 20 |

**Table 4. Mean, median and range of weaning completion age for different estimation methods as described in the analyses, compared by (a) sex and (b) region.**

**(a)**

| Estimation Method | Sex | Mean GLG | Median GLG | GLG Range (yrs) |
|---|---|---|---|---|
| A | F | 4.6 | 5 | 4–5 |
|  | M | 4.4 | 5 | 3–5 |
| B | F | 3.7 | 4 | 2–5 |
|  | M | 3.1 | 3 | 2–5 |
| C | F | 3.4 | 3 | 2–5 |
|  | M | 3.1 | 3 | 2–5 |

**(b)**

| Estimation Method | Region | Mean GLG | Median GLG | GLG Range (yrs) |
|---|---|---|---|---|
| A | Iceland | 4.3 | 4 | 4–5 |
|  | Labrador | 4.5 | 5 | 3–5 |
| B | Iceland | 4.0 | 4 | 3–5 |
|  | Labrador | 3.1 | 3 | 2–5 |
| C | Iceland | 2.7 | 3 | 2–3 |
|  | Labrador | 3.5 | 3 | 2–5 |

stomach contents for a single calf [22]. This new evidence of extended care in *H. ampullatus* has implications for the life history and energetics of other species of beaked whales, as well as their ability to recover from the effects of whaling or other population level impacts such as disease or mass stranding events due to mid-frequency active sonar (MFAS) [37].

While the nursing duration varies widely across mammal taxa, it is known to be generally related to maternal body size, as prolonged nursing helps fulfill the caloric requirements for growth of larger independent animals [3,38]. Weaning typically occurs when offspring reach a certain size, and while beaked whales have proportionally larger calves compared to other cetaceans [21], between birth and age five juvenile *H. ampullatus* almost double their length from three to six meters, with adult whales reaching 7–9 meters [20]. Although the calves of the largest odontocete, *P. macrocephalus*, are relatively smaller at birth, (~ 33% of maternal size), they have prolonged lactation and nursing (mean 36 months, range 2–13 years [25,39]), presumably to support their growth and development. Due to the large calf size of beaked whales and prior assumptions of their short nursing duration and inter-calf intervals, it has been suggested that their reproduction somewhat resembles the capital breeding energetics of baleen whales (e.g. [5,21]). Unlike beaked whales, however, baleen whales are bulk feeders able to ingest large amounts of food over short time periods [40], limited by life history attributes tied to the seasonal constraints of migration and ocean productivity, and have significantly higher average milk fat percent to support the rapid growth, development and weaning of their calves [3,38].

Although the composition of whale milk is poorly documented across species, odontocetes are generally known to have energetically less rich milk (mean fat = 24%) than baleen whales (mean fat = 33%) [3,38]. The only two records available for beaked whales suggest their milk fat % is even lower than average for odontocetes, based on single records of specimens of *H. ampullatus* (20%) and *Mesoplodon stejnegeri* (17%) [38]. However, milk energy output is not strictly based on fat composition, as solids (protein, sugars and ash or minerals) also contribute to total calories available for consumption. For the odontocetes where total milk energy output has been calculated (*P. macrocephalus*, *Kogia breviceps*, *Delphinus delphis*, [38]), it is notably low, comparable only to values found in primates, which are also known to have long lactations and extended periods of dependency. While data are not available to calculate the energetic output of *H. ampullatus* milk, similar to other medium to large odontocetes, we suggest

that prolonged nursing contributes to the caloric demands of rapid juvenile growth in the first 3 to 5 years.

Beyond providing necessary nutrition, nursing in mammals serves multiple functions; cetacean calves depend on nursing for their thermoregulation in the conversion of high fat milk into blubber, and maternal proximity offers protection from predators, ongoing socialization, and other important learning opportunities such as foraging and migration routes [3]. Prolonged nursing and gradual weaning, as part of the transition to nutritional independence, could be a life history adaptation for odontocetes with complex foraging strategies, such as deep diving. Both the biological demands and technical skills of foraging at depth may require time for physiological development and social learning. Although Newsome et al.'s study [18] of P. macrocephalus GLGs, found a gradual decrease in $\delta^{15}$N over the first 5 years, indicative of prolonged nursing, depth-recording tags indicated 1-year old calves had the capacity to dive to depths and durations of adult whales [41]. Whether H. ampullatus calves are also capable of diving to depths recorded for adult whales (e.g. 800–1400 m, [42]) is currently unknown. However, as juvenile beaked whales are overrepresented in mass stranding events linked to naval sonar, Hooker et al. [43] suggested that other aspects of dive capacity such as body mass, lung volume, or endurance for repeated dives, may be developmentally limited. We do know that for many species with a single precocial offspring, their young are introduced to solid food early despite prolonged nursing [3]. Thus, the need for prolonged maternal care in deep divers may also relate to the technical, socially learned aspects of foraging at depth, such as prey identification, capture and coordination with conspecifics.

While most isotopic studies of ontogeny have focussed on differences in $^{15}$N, here we also observed a regular pattern of increasing $\delta^{13}$C values from GLG 1 to older GLGs, which we suggest is consistent with weaning physiology. Milk is rich in $^{13}$C-depleted lipids, which if they are being incorporated into proteins, would lead to nursing animals having lower $\delta^{13}$C values than adults [12,44,45]. Although the trend for carbon is consistent with our inferences of prolonged nursing and a gradual transition from milk to solid food, gradual enrichment in $^{13}$C has not always been observed in other studies of odontocetes (e.g. D. leucas, [16]; Grampus griseus, [17]). As juvenile H. ampullatus whales learn to forage deeper, the increase in $\delta^{13}$C may reflect increasing consumption of bentho-pelagic species, which would be expected to have higher $\delta^{13}$C values [46]. Baseline $\delta^{13}$C can also vary spatially with latitude [47], and if all individuals demonstrated an ontogenetic shift in distribution it could potentially cause an increase or decrease in $\delta^{13}$C observed in tissues (e.g. [48]). However, based on global $^{13}$C isoscapes models [49], the lower latitudes ($< 40°$) where substantial foraging would have to occur to influence their $\delta^{13}$C profile, are at least 20° south of northern Labrador and Iceland, and outside of the known southern limit for the range of this species.

The patterns we observed appear largely consistent across a large number of specimens, however as a result of only including teeth with clearly defined GLG structure, we accept that our estimate of nursing duration may be biased towards healthy individuals. It is possible that age at weaning completion could be underestimated if the individuals in the study were weaned earlier due to available resources, or overestimated if maternal investment was longer than average. As our primary dataset included animals of different ages with a range of birth years spanning 1944–1967 (i.e. over four decades) it is unlikely either of these factors biased our results. The distinct GLG $\delta^{15}$N and $\delta^{13}$C patterns in the two whales which stranded in Newfoundland suggest that both individuals weaned earlier than the other specimens (at age 1–2, Fig 3a). Although there is no clear understanding of the relationship between $\delta^{13}$C and poor health conditions such as disease in whales, blubber stores may be mobilized during starvation or fasting (e.g Ursus americanus, [50]), and $^{13}$C depleted lipids would be incorporated in incremental tissues such as dentine. A notably decreasing rather than increasing $\delta^{13}$C trend

(Fig 3b) in the stranded male whale could reflect a longer period of physiological decline. Absolute isotopic values of carbon and nitrogen also suggest that the diet of Newfoundland whales was distinct from the specimens killed in Labrador and Iceland 30 years prior (Fig 5a and 5b). While we attempted to account for known climatic trends in $\delta^{13}$C (i.e. Suess effect) by adjusting our $\delta^{13}$C values, other temporal influences we cannot account for, such as other baseline isotope or other ecosystem shifts, may have occurred across the North Atlantic during the ~ 30-year period separating specimens. This highlights some of the challenges in using stranded animals of unknown health status and specimens from disparate time periods to make broad inferences on poorly understood species biology. Further investigation on the relationship between health status and the appearance of GLG structure in marine mammals would help clarify the influence of these factors for future studies.

Interestingly, our finding that $\delta^{15}$N in fetal dentine was almost 1 ‰ lower than GLG 1 differs from the pattern of steady decline in $\delta^{15}$N from a peak in fetal dentine observed in other species of cetaceans (*Grampus griseus*, [17]; *Monodon Monoceros*, Zhao *et al*. unpublished data; *Delphinapterus leucas*, Matthews & Zhao, unpublished data). Our explanations for the inconsistencies between enrichment patterns in fetal dentine across cetacean species consider two possibilities: (1) if tissues measured in other studies are actually neonatal rather than fetal dentine, $\delta^{15}$N for other species would reflect an ongoing decline in post-partum nursing [51]; or (2) differences are due to species-specific reproductive biology, such as physiological differences between capital and income breeders or growth dependent trophic enrichment factors. While occasional errors in identification of fetal dentine may occur, as Stewart & Stewart [52] describe there are multiple established landmarks for distinguishing pre and post-natal dentine deposition, making it unlikely that this is the source of consistent error across studies. Borrell *et al*. [12] found fetal tissues of capital breeders, which sustain reproduction with stored fat reserves, were higher in $\delta^{15}$N than their mothers, whereas for income breeders, mother-fetus $\delta^{15}$N discrimination was not observed. While odontocetes are generally recognized as income breeders, as per Huang *et al*. [21], aspects of *H. ampullatus* prenatal reproductive energetics, such as large relative calf size, do not align with the other odontocete species. Alternatively, if growth dependent $^{15}$N enrichment occurs due to rapid development in utero, it could explain fetal $\delta^{15}$N patterns, which may be different in smaller cetacean species than for larger species such as *H. ampullatus*. The inconsistencies in fetal development between species highlight the need to better understand the influence of maternal physiology on fetal development and stable isotope discrimination so that future studies can accurately interpret stable isotope profiles [12].

Theory predicts that parents in polygynous species may adopt a sex-bias in infant investment towards males [53]. While we do not have a good understanding of the mating systems across any of the species of beaked whales [54], most are sexually dimorphic, and in *H. ampullatus*, males are significantly larger in size, suggesting they need additional energetic resources for growth [55]. Although Hooker *et al*. [56] found adult males were marginally enriched in $^{15}$N relative to females, we did not find significant evidence that this occurs as part of maternal investment. While there may be some influence of sex on trophic position in mature animals, there was no difference between males and females in terms of nursing duration, or relative values of $\delta^{15}$N or $\delta^{13}$C across GLGs. However individual variation and annual averaging within GLGs may mask the presence of finer scale sex-based patterns or trends in isotopic enrichment (Figs 3a and 4) [57].

The weaning period, which includes the introduction to solid food accompanied by nursing, can vary in length depending on whether maternal weaning strategies are abrupt or gradual. Using the timing associated with the cessation of a general declining trend in $\delta^{15}$N, changepoint analysis or model fit against a number of theoretical curves, a number of authors

[14,16,17] have proposed that unlike baleen whales, weaning in odontocetes is a relatively gradual process. While the introduction of solid food may occur within the first year, this is often accompanied by prolonged nursing across a number of odontocete species [4], suggesting stomach contents are unlikely to provide good evidence of the age when weaning is complete. The point when $\delta^{15}N$ values become relatively stable and more consistent with subsequent GLGs or are approximately equivalent to mature baseline values has been used to estimate weaning completion [14,16,51]. For *H. ampullatus*, we found generally similar results across methods, suggesting weaning was complete when whales were between 3 to 4 years old. The point when $\delta^{15}N$ values stopped decreasing (Method A), suggested $\delta^{15}N$ declined into year five for some individuals, which could reflect individual variation in prolonged nursing, or differences in ability to forage on higher trophic level prey. Defining weaning completion as the point when $\delta^{15}N$ was equal to mature values (Method B), or when $\delta^{15}N$ was 1.2‰ lower than GLG1 (Method C), suggests that in *H. ampullatus*, similar to other odontocetes, nursing is prolonged with weaning taking over 3 years to complete.

If Benjaminsen [22] was correct in their calculation of a 12 month gestation period for *H. ampullatus*, nursing a calf for at least 3 years would double previous estimates of their reproductive cycle to at least 4 years [20]. New *et al.*'s [5] bioenergetic models of beaked whales found that low survival and reproduction was tied to the relatively short estimates for duration of lactation, and the assumption of a 2-year calving interval. Energetically, a large percentage of beaked whales in New *et al.*'s [5] models had difficulty meeting their metabolic requirements under standard assumptions and inferred reproductive parameters derived from historic whaling data. Prolonged nursing was identified by New *et al.* [5] as an alternate strategy that would give females a recovery period between mating, allowing them to rebuild energetic stores and increase the probability of their next calf's survival. Prolonged maternal investment and a longer inter-calving interval also has consequences for the rate of effective population growth. Given the assumption that for most odontocetes, pregnancy and lactation rarely overlap, extended nursing decreases the lifetime reproductive potential of the species by half. For *H. ampullatus*, extended maternal care would prolong their recovery from commercial whaling and increase the impact of contemporary risks to their populations such as disease outbreaks, MFAS induced strandings or other unusual mortality events [3–5,37,58,59].

A longer nursing period also implies that *H. ampullatus* have extended maternal associations, and suggests that social structure of beaked whales may be more complex than previous observational studies have been able to detect [54,60]. Generally beaked whales are found in very small groups and are not considered particularly social, however in well studied beaked whales (e.g. *Ziphius cavirostris*, *Berardius sp.*, *Mesoplodon densirostris)*, there is some evidence of long-term bonds (over months to years) between individuals using photo-ID methods [19,54]. Although McSweeney [61] documented repeated associations over two years between a female *Ziphius cavirostris* and her calf, and Baird [54] suggests that *M. densirostris* calves disperse from their mothers between 2–3 years of age, long-term associations with relatively unmarked beaked whale calves are particularly hard to track using photo-identification. In the only study where putative mother-calf relationships were assessed in *H. ampullatus*, repeated associations over two subsequent years were only documented twice [60,62]. From our review, the range of estimates for the duration of lactation, weaning period, age of dispersal or inter-calf interval in beaked whales has either been inferred from the maximum length of maternal-calf associations using photo-identification analysis or applied across species using limited stomach content data (e.g. see [5]). Thus, our study provides the first significant dataset for interpreting the range of variation in individual maternal investment in a species of beaked whale and improves our understanding of the diversity in maternal strategies found across cetaceans and mammals.

## Supporting information

**S1 Table. Summary of cetacean studies reporting nursing duration or weaning age by species, method, average age at weaning, and sample type.**
(DOCX)

**S2 Table. Rationale for inclusion of variables and data sources.**
(DOCX)

## Acknowledgments

We would like to acknowledge the contributions of the GG Hatch Lab to the analysis of these samples, and support of Dr. Steve Ferguson, DFO. Teeth specimens from whaled animals were contributed by Nils Øien, Institute of Marine Research Norway. Samples from stranded whales were generously contributed by Wayne Ledwell, of the Newfoundland Stranding network.

## Author Contributions

**Conceptualization:** Laura Joan Feyrer, Cory J. D. Matthews.

**Data curation:** Laura Joan Feyrer, Shu ting Zhao.

**Formal analysis:** Laura Joan Feyrer, Shu ting Zhao, Hal Whitehead, Cory J. D. Matthews.

**Funding acquisition:** Laura Joan Feyrer, Hal Whitehead, Cory J. D. Matthews.

**Investigation:** Laura Joan Feyrer, Cory J. D. Matthews.

**Methodology:** Laura Joan Feyrer, Shu ting Zhao, Hal Whitehead, Cory J. D. Matthews.

**Project administration:** Laura Joan Feyrer, Cory J. D. Matthews.

**Resources:** Laura Joan Feyrer, Cory J. D. Matthews.

**Supervision:** Hal Whitehead, Cory J. D. Matthews.

**Validation:** Laura Joan Feyrer, Shu ting Zhao, Cory J. D. Matthews.

**Visualization:** Laura Joan Feyrer.

**Writing – original draft:** Laura Joan Feyrer, Cory J. D. Matthews.

**Writing – review & editing:** Laura Joan Feyrer, Shu ting Zhao, Hal Whitehead, Cory J. D. Matthews.

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
