## [Decision Letter · Decision Letter 0]

13 May 2020

PONE-D-20-09337

Prolonged maternal investment alters our understanding of beaked whale reproductive life history

PLOS ONE

Dear Ms Feyrer,

Thank you for submitting your manuscript to PLOS ONE. After careful consideration, we feel that it has merit but does not fully meet PLOS ONE’s publication criteria as it currently stands. Therefore, we invite you to submit a revised version of the manuscript that addresses the points raised during the review process.  In addition to the reviewer comments and editorial suggestions, please address these particular changes: 

- Clarify the sample sizes: In the introduction (line 105), 151 teeth are noted (from 151 individuals?), while in the methods (line 126 and 130), 50 is noted (from 50 individuals?), as on line 144 ('only 50'). Later (line 184) this 50 is reduced to a sub-set of 39, although it is unclear why the 50 with GLGs1-5 were reduced to those with GLGs1-3? Please examine the original sample (n = 151 individuals) and walk the readers through the steps taken to reach the final (n = 50 teeth) sample size.

- Similarly, the discussion (line 302) notes "across 50 individuals", but based on Figure 3 it seems this should be 39? In Figure 4 the sample size values for males and females should be explained in the caption, as these could easily be construed as the number of individuals (which presumably they are not).

- In Table 1, please clarify the following numbers: n=288/244, n=207, n=124, n=112, n=110, n=97

- In Figure 4 a-b and 5a-b. Report the sample size for each point, not the total.

- In Table 2. Sort based on increasing AIC values.  Cannot see results for null model in your candidate models?

- You should include it to your list: d15N~1+(1|Tooth_N).

- In Table 3. Modify italic font to bold on the first line of Table 3a.

- In addition, modify 0.000 entries by >0.001 for very small p-values.

- In Lines 287-288 and Table 4.  Be consistent with number of decimal places shown for these values.

- In Lines 113, 320, 333, 337, 367, 404. 

Make sure the use of the common and scientific names is consistent.

Currently, sometimes they are both used, sometimes only one or the other are used, …

- Line 470. This discussion could be broadened to include population recovery in response to other impacts (e.g., MFAS, disease outbreaks).

We would appreciate receiving your revised manuscript by Jun 27 2020 11:59PM. To enhance the reproducibility of your results, we recommend that if applicable you deposit your laboratory protocols in protocols.io, where a protocol can be assigned its own identifier (DOI) such that it can be cited independently in the future. For instructions see: http://journals.plos.org/plosone/s/submission-guidelines#loc-laboratory-protocols

We look forward to receiving your revised manuscript.

Kind regards,

David Hyrenbach, Ph.D.

Academic Editor

PLOS ONE

Journal Requirements:

2. In your Data Availability statement, you have not specified where the minimal data set underlying the results described in your manuscript can be found; the DOI address provided does not show data.

PLOS defines a study's minimal data set as the underlying data used to reach the conclusions drawn in the manuscript and any additional data required to replicate the reported study findings in their entirety. All PLOS journals require that the minimal data set be made fully available. For more information about our data policy, please see http://journals.plos.org/plosone/s/data-availability.

3. We note that Figure 1 in your submission contains map/satellite images which may be copyrighted.

We require you to either (a) present written permission from the copyright holder to publish this figure specifically under the CC BY 4.0 license, or (b) remove the figure from your submission:

b. If you are unable to obtain permission from the original copyright holder to publish this figure under the CC BY 4.0 license or if the copyright holder’s requirements are incompatible with the CC BY 4.0 license, please either i) remove the figure or ii) supply a replacement figure that complies with the CC BY 4.0 license. Please check copyright information on all replacement figures and update the figure caption with source information. If applicable, please specify in the figure caption text when a figure is similar but not identical to the original image and is therefore for illustrative purposes only.

Reviewers' comments:

Reviewer's Responses to Questions

**Comments to the Author**

1. Is the manuscript technically sound, and do the data support the conclusions?

Reviewer #1: Yes

Reviewer #2: Yes

2. Has the statistical analysis been performed appropriately and rigorously? 

Reviewer #1: Yes

Reviewer #2: Yes

3. Have the authors made all data underlying the findings in their manuscript fully available?

Reviewer #1: Yes

Reviewer #2: No

4. Is the manuscript presented in an intelligible fashion and written in standard English?

Reviewer #1: Yes

Reviewer #2: Yes

5. Review Comments to the Author

Reviewer #1: Reviewer comments

General comments:

The authors study the nursing and weaning periods in the northern Bottlenose whale according to stable isotope values of nitrogen and carbon. They show an extended nursing period (3 to 5 yrs) in comparison to what was previously thought in the species (1 yr). This finding helps to better understand the entire reproductive cycle of the species. This is a very interesting study providing useful fundamental knowledge on the biology of a cryptic species. I have some minor suggestions to improve the manuscript which is overall very well-written with a robust statistical design. I suggest this manuscript should be accepted after minor revisions.

Specific comments:

l.182: change the font of <delta>

l.184: n=39 or n=37 as appears in Table 1?

Table 1: n=288/244, n=207, n=124, n=112, n=110, n=97: what are those numbers? Could you please specify?

Fig 4a-b and 5a-b: You should give the sample size for each point, not the total. Otherwise it is confusing.

Table 2: I do not see the results for the null model in your candidate models? You should include it to your list: d15N~1+(1|Tooth_N).

Table 3: Modify italic font to bold on the first line of Table 3a. In addition, you should modify 0.000 entries by >0.001 for very small p-values.

Reviewer #2: Overall this paper is well-written and appears technically sounds. I have a number of mainly minor comments. I did not see a Data Availability Statement with the manuscript.

Beaked whales are a speciose group, and this study focuses on only one species. I'd suggest either discussing what is known of weaning age in other well-studied beaked whales (Berardius, Ziphius, M. densirostris) or narrowing the focus on the title to "... northern bottlenose whale reproductive life history". For species that do not have long-term bonds (e.g., Ziphius, M. densirostris) the age of dispersal of an offspring away from its mother would also be a good indicator of maximum age of weaning. It would be good to note whether any information is available on this from other species of beaked whales that have been the subject of photo-ID studies. Some reference (in lines 304-308) to what is known (or not known) about northern bottlenose whale disassociations between mothers and calves based on photo-ID would be warranted.

Line 27, 136, 483. The term "whaled" is used instead of "hunted". Would be more explicit to use hunted.

Line 43. Use "factors" instead of "constraints"?

Lines 100-103. The sentence "However beaked whales are...." is not particularly relevant and not a good transition sentence for introducing the purpose of this study. That they only have two teeth or that the teeth only erupt in adult males doesn't fit with the general issue of ontogenetic diet shifts.

Sample size. In the introduction (line 105), 151 teeth are noted (presumably from 151 individuals?), while in the methods (line 126 and 130), 50 is noted (presumably from 50 individuals?), as on line 144 ('only 50'). Later (line 184) this 50 is reduced to a sub-set of 39, although it is unclear why the 50 with GLGs1-5 were reduced to those with GLGs1-3? Either way, would be good to bring up the original sample (151 individuals) examined and walk through the steps getting to the final sample. Similarly, the discussion (line 302) notes "across 50 individuals", but based on Figure 3 it seems this should be 39? In Figure 4 the sample size values for males and females should be explained in the caption, as these could easily be construed as the number of individuals (which presumably they are not).

Table 2. Good to sort this based on increasing AIC values.

Table 3. Why is the 0.005 value in italics? For those values listed as 0.000 better to list as <0.001.

Line 287-288 and Table 4. Good to be consistent with the number of decimal places shown for these values.

Lines 113, 320, 333, 337, 367, 404. I'm not sure what the formatting requirements are for PLoS ONE but the use of common and scientific names is inconsistent (sometimes both, sometimes only one or the other), and would be good to be consistent.

Line 470. This could be broadened to include population recovery in response to other sources of decline (e.g., MFAS, disease outbreaks).

  </delta>

6. PLOS authors have the option to publish the peer review history of their article (what does this mean?). If published, this will include your full peer review and any attached files.

Reviewer #1: Yes: Sabrina Tartu

Reviewer #2: Yes: Robin W Baird

---

## [Author Response · Author response to Decision Letter 0]

3 Jun 2020

Thank you for the helpful comments provided by the editor and reviewers. We have made all the suggested changes as requested. Please see attached submision for detailed responses and line numbers.

---

## [Editor Report · Decision Letter 1]

10 Jun 2020

Prolonged maternal investment in northern bottlenose whales alters our understanding of beaked whale reproductive life history

PONE-D-20-09337R1

Dear Dr. Feyrer,

We’re pleased to inform you that your manuscript has been judged scientifically suitable for publication and will be formally accepted for publication once it meets all outstanding technical requirements.  Thank you for carefully addressing the comments from tha reviewers and the editor, and implementing the necessary changes.

Kind regards,

David Hyrenbach, Ph.D.

Academic Editor

PLOS ONE
---

## [Editor Report · Acceptance letter]

12 Jun 2020

PONE-D-20-09337R1 

Prolonged maternal investment in northern bottlenose whales alters our understanding of beaked whale reproductive life history 

Dear Dr. Feyrer:

I'm pleased to inform you that your manuscript has been deemed suitable for publication in PLOS ONE. Congratulations! Your manuscript is now with our production department. 

Kind regards, 

on behalf of

Dr. David Hyrenbach 

Academic Editor

PLOS ONE